# Institutional Network Relationship of Chinese Public Crisis Governance System—Based on the Quantitative Comparative Analysis of Policies during SARS and COVID-19

**DOI:** 10.3390/ijerph192215100

**Published:** 2022-11-16

**Authors:** Jian Yang, Weikun Huang

**Affiliations:** School of Management, Anhui University, Hefei 230039, China

**Keywords:** public crisis governance system, policy document metrology, SARS, COVID-19, crisis management

## Abstract

After experiencing many public crisis events, such as SARS in 2003 and COVID-19 in 2020, the Chinese public crisis governance system has been improved from its initial state. The distribution structure and cooperation network among various government departments in China have become more complex. How to accurately clarify the relationship between the various departments in the existing governance system has become an important issue of the Chinese public crisis governance system. Based on the perspective of networked research, this article examines the network relationship between institutions in the Chinese public crisis governance system from the two dimensions of network centrality and network density. Using the bibliometric method to use public policies released in 2003 and 2020 as data samples and the two large-scale institutional reforms in 2003 and 2018 as the time nodes, this paper conducts a comparative analysis of the institutional network relationship of the Chinese public crisis governance system during different periods. The research shows that the network relationship among institutions in the Chinese public crisis governance system has changed from a centralized type to a diverse type; there is a trend of expansion in network relations; the legalization of governance networks is strengthened and the core of the network is transformed into the direct leadership of the Party Committee; and the overall network structure is experiencing a rational evolution.

## 1. Introduction

Since the SARS outbreak in 2003, the Chinese government and a large number of scholars have begun to reflect and pay attention to the construction of crisis management systems. How to deal with and improve the crisis management mechanism and organization construction has gradually become a question that public management scholars need to think about and answer.

As early as 1955, the Ministry of Health promulgated “the Measures for the Administration of Infectious Diseases”, and in 1989, it promulgated the first “Law on the Prevention and Control of Infectious Diseases” in China. During the fight against SARS, “the Regulations on Emergency Response to Public Health Emergencies” and the revised “Law on Prevention and Control of Infectious Diseases” were promulgated and implemented in due course. Thereafter, “The National Emergency Plan for Public Health Emergencies” and “the Emergency Response Law” have been promulgated and implemented one after another. In the meantime, the Chinese emergency management system has been rapidly improved, and 16 deliberation and coordination institutions of the State Council related to emergency management have been formed, as have emergency management offices covering all provincial governments and most municipal and county governments. In 2004, China built and started the world’s largest infectious disease surveillance network. In 2018, the Ministry of Emergency Management was set up, and the National Health and Wellness Committee was adjusted. After the Fourth Plenary Session of the 19th Central Committee in 2019, the organizational reshaping and functional recasting under the goal of modernization of the Party’s leadership governance were accelerated [1].

In the face of public crisis, which is a battlefield without smoke, strengthening the innovation of crisis management and the control mechanism, integrating information strategic resources, and stimulating citizens’ sense of social responsibility are the only ways through which the perfect combination of technical defense and civil air defense in the “battle” be realized and a more systematic, complete, scientific, standardized, and effective crisis management system be constructed, promoting the modernization of the Chinese governance system and governance capacity [2].

Leadership is one of the major elements in the crisis management system, and the construction of social disaster management institutions can improve the national crisis and emergency management [3]. The nature of national leadership and the organization of government are factors that helped to shape policy responses to crisis [4]. Christensen and Laegreid believe that a reliable and professional bureaucracy is one of the reasons why Norway has performed well in handling the coronavirus pandemic [5]. Because of the complexity and intersection of public crisis governance, governance institutions need to transcend their own responsibility boundaries, break through the “clear division of labor” emphasized by the traditional bureaucratic theory, and carry out frequent and in-depth cooperation [6]. Based on the archives and interview data, Liu analyzed the changes in the crisis management structure in China and believes that both vertical and horizontal coordination of the crisis management institutions became stronger over time [7]. At present, scholars’ research on the public crisis governance system basically covers theoretical explorations of the overall governance system, that is, how to reform and improve the crisis governance system, or theoretical explorations of different system frameworks, that is, putting forward various more reasonable governance systems.

The search results of Web of Science show that 6895 studies in the literature have been searched on the theme of “public crisis” or “crisis management”; with the theme of “governance institution”, a total of 604 studies were searched. In terms of quantity, both topics have been studied by many scholars. However, only 65 studies have been searched on the theme of “institutional relationship”, and in the search results for “public crisis” or “crisis management”, the search result of “institutional relationship” shows that there are no studies. It can be seen that few scholars have conducted in-depth research on the institutional relationship, especially in the public crisis governance system.

Accurately grasping the distribution structure of public crisis governance institutions and clarifying the cooperation network structure in different periods have academic value and practical significance for the modernization of Chinese public crisis governance. The interaction and cooperation between Chinese government agencies on public crisis management has been developed with the reform of government agencies and the occurrence of specific events. Since the SARS epidemic in 2003, the Chinese government has begun to pay attention to the institutional reform of the public crisis management system. Before the COVID-19 epidemic, it also experienced a series of local public health events. Accordingly, the ability of the Chinese public crisis management system is gradually improving, and the full-scale outbreak of the COVID-19 epidemic is just a test of governance ability. Therefore, this paper chooses the SARS and COVID-19 epidemics as the research focus to compare and analyze the internal changes in the mechanism of the inter-agency network relationship in the Chinese public crisis management system. Taking two large-scale institutional reforms in 2003 and 2018 as the time nodes, this paper investigates the structure and evolutionary characteristics of the cooperation network between institutions belonging to the Chinese public crisis governance system at two important time nodes. This paper aims to enrich the knowledge of the relationship between Chinese governance institutions in the governance of major public health events. It also hopes to provide reference for other crisis governance in China and public health governance in other countries.

## 2. Determination of Basic Concepts

As the research theme of this paper, the inter-agency relations of a public crisis governance system include two core concepts: the public crisis governance system and inter-agency relations. However, because different scholars explain the concepts from different perspectives, the connotations of the concepts are inconsistent in different research. In order to aid the accuracy of this study, this paper defines the concepts in this section.

### 2.1. Public Crisis Governance System

Most of the research on public crisis management by Chinese scholars follows the principle of multi-agent cooperation, including cooperation between governments, between governments and non-governmental organizations, between governments and social organizations, and so on. Among them, Liu Xia believes that the Chinese public crisis governance system is a highly open organizational network with the government as its lead, with the third sector participating in the organization and the enterprises cooperating institutionally and interacting from top to bottom [8]. Transforming crisis management from management to a governance system is a prerequisite for realizing the value of crisis governance. This is achieved through a system for the efficient division of labor with clear responsibilities to effectively achieve the government’s goals and to improve the government’s efficiency, transparency, responsibility, and credibility. The governance system mentioned here has a flexible network organization structure that is far superior to the traditional bureaucratic management system in terms of timeliness and flexibility in the face of an unknown crisis. Flexibility and adaptation are key assets of a well-functioning governmental crisis management system [9]. Therefore, this kind of network system has the best organizational form and is conducive to equal cooperation, joint decision making, and concerted action. In this network structure, various organizations, departments, and even individuals or other subjects can cooperate with each other anytime and anywhere, thus realizing the timeliness and convenience of management.

Chang Mingjie [10], Wang Lin [11], Zhang Qin [12], and other scholars all believe that the Chinese public crisis governance system needs to transform from the unitary government subject mode and the concentric circle governance mode of “center-edge” to a network cooperation governance mode. The network mentioned here is an English-language network. Instead of the Internet, what we call the “network” here refers to a comprehensive multi-agency cooperation network with relatively close links among institutions with obvious boundaries; great flexibility as well as concentration; and equal and symmetrical relative status, resources, and information. Based on this, we call this cooperative network system on which governance depends the governance network system.

### 2.2. Inter-Agency Relations

“At the heart of understanding how governments respond to crises are notions of bureaucratic coordination.” Ref. [13]. Bureaucratic coordination must involve the inter-agency relationships. The definition of inter-agency relationships has always been vague, and some scholars believe that only government agencies can be called institutions [14]. Others believe that institutions not only refer to government agencies, but also include non-governmental organizations in civil society. For example, non-governmental organizations; societies organized by citizens spontaneously; and associations, community organizations, and various movements organized by citizens spontaneously are also called third-sector and civil society organizations [15].

Wei, Fan, and Meng, defined inter-agency relationships as formal or informal interactions and game mechanisms formed by different governance institutions during the process of public affairs governance [16]. That is, inter-agency relations not only include inter-governmental relations within the system, but also include the interaction between institutions outside the system and institutions inside the system. For example, Chen developed a “central government-local government-society” framework to analyze public crisis management during the COVID-19 pandemic to reveal the key features of government–CSOs interactions in China [17]. By comparing the responses of China and South Korea to COVID-19, Mao emphasizes that state capacities are shaped by the central–local government relations and state–society relations [18].

According to the above analysis, due to the operability of policy measurement and the high concentration of policies in the face of major public crisis, this paper defines inter-agency relationships as the formal or informal interaction and game mechanism of government agencies at the central level and some mass organizations and trade associations involved in public crisis governance.

## 3. Research Methods and Data Basis

Analyzing the relationship structure of the public crisis governance institutions is a complex process because of the interaction between institutions. It requires a systematic research method to reflect the cooperative network relationship between institutions. Therefore, this study selects the social network analysis (SNA) method to carry out precise quantitative analysis of various relationships between institutions. In this study, taking the existing policy text as the data sample, it not only facilitates the acquisition of samples, but also guarantees the comprehensiveness.

### 3.1. Research Methods

In recent years, Chinese scholars have conducted in-depth research on various disciplines through social network analysis. For example, by analyzing the social network of scientific and technological cooperation in countries along The Belt and Road Initiative, Chen obtained the overall network structure, core nodes, topological structure, and the evolution process of high-frequency cooperation pairs [19]. Zhu and Cheng, by analyzing the social cooperation network of the policy subjects of the transformation of production, education, and research achievements in China, reached the conclusion that the network scale is expanded, and the influence of core nodes is enhanced [20]. Gao and Chen studied the dynamic mechanism and structural evolution characteristics of the formation of an industry–university–research cooperation innovation network in the ICT field in China [21]. It can be seen that the social network analysis method has been widely used in research on various subject relationship networks. With the continuous development of society, unknown crises are becoming more and more complex, so it is becoming more and more important to use social network analysis methods to investigate and analyze the relationship between institutions belonging to the public crisis governance system.

At present, most of this research comprises theoretical studies, and there are few quantitative studies on its binding correlation. The reason why the research on inter-agency relations is mainly qualitative and normative is that it is difficult to obtain high-quality social relations data [22]. In the public crisis governance policy system, “joint writing” between different governance institutions is a relatively special type of policy. The “joint writing” not only reflects the complexity and intersection of the public crisis governance field, but also reflects the complex and subtle institutional relations under the Chinese special political system and culture, providing a direct perspective for the study of cooperation and interaction between public crisis governance institutions. By extracting structural elements (publishing institutions) from public policy texts, the quantitative analysis of the inter-agency structure of the Chinese public crisis system is realized through a comparative analysis of the number of joint publications and the number of independent publications of each institution.

Based on the above analysis, this paper regards the joint publishing behavior of different institutions for the same crisis event as the existence of a liaison relationship between two institutions and constructs a network system, calculates the density and centrality of different network relationships using Ucinet6, and draws an inter-agency relationship network map of the Chinese public crisis governance system using NetDraw. The reason why density and centrality are used to describe the inter-agency relationship network map of the Chinese public crisis management system is that centrality and density can more accurately describe the morphological characteristics of the network than other indicators [23].

The above-mentioned centrality and density are calculated using the Ucinet6 software. Both of them are quantitative descriptions of the overall network structure. In more detail, the former is more inclined to describe the concentration degree of a network structure. For a certain node, centrality indicates that this node is in the position of the overall network structure, and the greater the centrality, the higher the position of the organization at this node in the network system. Accordingly, the concept of centrality is extended to the whole network system, and centrality evolves into central potential. The greater the central potential, the more centralized the whole network structure and the higher the degree of centralization. On the contrary, lower central potential means that the whole network structure is controlled by multiple nodes and that the network structure system is more diversified. The latter tends to describe the degree of association between nodes in a network structure. The higher the density, the closer the connection between nodes in this network structure system. The smaller the density, the smaller the proportion of interconnection among nodes in the network structure; almost all of them are in an independent state, and the whole network is in a decentralized state.

### 3.2. Data Sources

These two epidemics are highly representative, and their long duration, wide range of influence, and strong transmission are all reasons why they have become representative public crisis events. Therefore, according to the main time periods of the SARS and COVID-19 epidemics, this paper collects relevant policies for epidemics from the Peking University Fabao Database. On this basis, the data are compared and supplemented with the websites of the central government, the National Health Commission, the Ministry of Transport, and other departments. Additionally, the following three requirements were included for screening: 1. data must be highly related to the SARS and COVID-19 epidemics, and the policies with low relevance should be removed; 2. data must be based on the policies promulgated by the State Council and the relevant departments of the State Council while also screening out local policies; and 3. certain policy texts with specific operational plans, such as relevant laws and regulations, measures, notices, opinions, circulars, etc., are selected, and cases, replies, trial plans, etc., are removed. In total, 637 epidemic-related policy documents released by the central government from 2 April 2003 to 30 December 2003 and from 23 January 2020 to 10 November 2020 are taken as research samples.

## 4. Comparative Empirical Analysis of Inter-Agency Network Relationship in Public Crisis Governance System

This section takes 637 policy texts published during SARS in 2003 and COVID-19 in 2020 as data samples and compares them according to the issuing level, issuing agency, and number of documents issued. Through the social network analysis method, the ucinet6 software was used to obtain the network centrality, network density, and other relevant data of the governance system in the two periods. The network atlas drawn by NetDraw software was used to intuitively analyze the crisis governance architecture in the two periods.

### 4.1. Descriptive Analysis of Policy Text

Among the 637 policy texts selected in this paper, the number of policy texts issued by an independent institution is 502, and the number of policy texts jointly issued by two or more institutions is 135. Among them, there were 182 policy documents in the SARS period and 455 policy documents in the COVID-19 period. During these two periods, the number of policy documents and issuing agencies is obviously unbalanced (see Table 1 and Table 2).

Among them, Table 1 shows the statistics of policy-issuing institutions during SARS. Table 1 shows that the number of departmental normative documents and departmental working documents issued by the departments of the CPC Central Committee, ministries and commissions under the State Council, directly affiliated institutions, and deliberation and coordination institutions is the largest, reaching 91.6%. Although the number of administrative regulations and normative documents issued by the State Council is small, these documents are high-level programmatic documents, which guide the basic direction of policy objectives during a public crisis. The judicial interpretations issued by the Supreme People’s Court and the Supreme People’s Procuratorate are relatively few, with only one document. Among the 182 policy documents, 5 documents are related group regulations and industry regulations issued by group organizations and industry associations.

Table 2 shows the statistics of policy-issuing agencies during the period of COVID-19. Similar to Table 1, Table 2 shows that the largest number of departmental normative documents and departmental working documents was issued by departments of the CPC Central Committee, ministries and commissions of the State Council, directly affiliated institutions, and deliberative and coordinating institutions, accounting for 90.8% of the total number of issued documents. As in the SARS period, the administrative regulations and normative documents issued by the State Council are also programmatic documents that are of a higher level and fewer in quantity. The number of legal documents issued by the Standing Committee of the National People’s Congress, the supreme law, the judicial interpretations issued by the Supreme People’s Procuratorate, and working documents is also small, with a total of nine documents. In total, seven documents comprise group regulations and industry regulations issued by group organizations and industry associations.

In contrast, the changes made by the Chinese government during the COVID-19 epidemic were more obvious than those made during the SARS epidemic. First of all, the number of policies has been greatly improved, which shows that the government’s response to public crisis is more comprehensive and meticulous. Secondly, regarding the proportion of joint publications in the total number of publications, the proportion increased greatly during the COVID-19 epidemic, from 12.7% to 24.6%. This shows that the linkage network of government departments is closer when dealing with public crisis.

### 4.2. Comparative Analysis of Inter-Agency Networks of Public Crisis Governance System

The text data were processed by Ucinet6, and the central potential and density of cooperation networks of different institutions in different time periods were calculated. Furthermore, NetDraw software was used to deal with the matrix relationship between different publishing organizations, and the network maps between different governance organizations at the central level during SARS and COVID-19 were drawn (Figure 1 and Figure 2). Each individual box in two network maps represents an independent governance institution, and the connection between two governance institutions means that there is joint publishing behavior between the two institutions. A thicker connection means that two institutions are more closely connected; that is, in cases where there is more joint publishing, the size of each box is determined by intermediary centrality. This means that the higher the number of shortest paths passing through a node, the larger the box and the more frequent the connection between this institution and other institutions is, and this plays an intermediary role in the whole network structure. In this paper, by drawing and calculating the network map and network central potential during the two epidemics as well as by combining the reform of government institutions in 2003 and 2018, we can conclude the characteristics and changes in network relations among institutions in the Chinese public crisis governance system.

#### 4.2.1. Inter-Agency Cooperation Network for Public Policy Making during SARS

According to Figure 1 and Table 1, there were 182 policies issued during SARS, of which 159 were issued separately and 23 were issued jointly. This epidemic can be described as the first infectious epidemic to have great influence in China. When the policy departments responded to this epidemic, the initial response speed was slow, and with the development of the epidemic, they gradually responded accordingly. Therefore, it can be seen that the Ministry of Health (which has renamed as the “National Health Commission”) occupies a central position in the policy release organization of this epidemic situation. The main reason is that most of the policies in this epidemic were introduced to curb the spread of the epidemic. The institutions that jointly issued documents were the Ministry of Finance, the National Development and Reform Commission, the Ministry of Civil Affairs, and the Ministry of Communications (which has been revoked). It can be seen that the goal of these departments is to maintain social stability and protect people’s livelihood.

During SARS, the inter-agency cooperation network for public policy making had a density of 0.063 and a central potential of 0.424, making it relatively low in density and relatively high in central potential. The main reason for this was that the Chinese government was facing the public crisis of a major epidemic for the first time at that time. In the face of this epidemic, the Chinese government adopted a highly centralized management mode. The linkage between various departments and institutions is relatively weak.

#### 4.2.2. Inter-Agency Cooperation Network for Public Policy Making during COVID-19

Compared to the status during SARS, the number of policy documents highly related to the epidemic situation during COVID-19 increased greatly, with a total of 455 policy documents issued, 343 of which were issued separately and 112 of which were jointly issued. This is not only related to the spreading ability and duration of the epidemic, but also is related to the globalization of the epidemic. How to prevent and control the epidemic overseas under normal conditions has become the focus of epidemic prevention and control. Therefore, the Civil Aviation Administration of China, the State Administration of Immigration, and the Ministry of Foreign Affairs have larger nodes in the map.

From the network map, it can be seen that the National Health Commission occupies the most dominant position, followed by the Ministry of Finance and the Ministry of Human Resources and Social Security. In this epidemic, the National Health Commission, which integrated the functions of the former Ministry of Health and the National Health and Family Planning Commission, played a leading role. In the initial stage of the epidemic, all of society was in a state of wartime: medical resources were extremely scarce, and people’s moods were unstable. At this time, it was necessary to adopt a bureaucratic governance method, with a general department planning resources uniformly, step by step, and keeping strict rules, so that resources could be used reasonably and fully. Under the influence of the big environment, people gradually became emotionally stable and consciously participated in epidemic prevention work.

It is important to emphasize that there is an institutional node in the map, the “Working Mechanism for Joint Prevention and Control of Pneumonia Epidemic in COVID-19”. Although this special institution does not have any connection with other institution nodes in the map, it does not mean that it is only a single issuing institution. It was established under the leadership of the National Health Commission. There are 32 departments in the member units. Under the joint prevention and control mechanism, there are working groups such as epidemic prevention and control, medical treatment, scientific research, publicity, foreign affairs, logistics support, and front work, which are headed by responsible officials of relevant ministries and commissions who have clear responsibilities and divisions of labor and cooperation to form an effective joint force for epidemic prevention and control [24]. During the epidemic period, the joint prevention and control mechanism issued a total of 78 policies. Among them, there are 11 normative documents of the State Council, 9 departmental working documents, and 58 departmental normative documents. From the level of issuing agencies, it can be seen that this mechanism involves many departments and has high flexibility, which meets the requirement for establishing a rapid response mechanism for public crisis management [25]. Decisive measures can be taken after the crisis through the use of various social resources, controlling the spread of the situation in time.

During the period of COVID-19, the density of the inter-agency cooperation network in public policy making was 0.098, and the central potential was 0.392. Compared with the SARS period, the network density is relatively high, and the network center potential is relatively low. This mainly shows that the policies issued by the Chinese government in the face of sudden epidemics are relatively flexible, the links between departments are becoming closer, and the centralized nature is relatively weakened. Regardless of the policy coverage or the government’s response speed to the epidemic, the linkage and cooperation between various departments have been greatly improved. Therefore, it can be seen that the inter-agency network relationship of the Chinese public crisis governance system is improving and becoming reasonable.

## 5. Discussion

According to the above analysis, the inter-agency network relationships of the Chinese public crisis governance system during the periods of SARS and COVID-19 are presented. By analyzing the inter-agency relationship, this section discusses the characteristics of the inter-agency network structure.

### 5.1. The Type of Inter-Agency Network Structure of the Chinese Public Crisis Governance System Has Changed from Centralized and Decentralized to Multivariate and Compact

According to the network central potential and network density, this paper constructs a two-dimensional model diagram that can divide the inter-agency network structure types of the Chinese public crisis governance system into four categories. As shown in Figure 3, the first category is multi-decentralized, which is characterized by low network centrality and network density, that is, low centralization and connection between institutions. The second category is centralized and decentralized. Its performance is characterized by high network centrality and low network density, that is, high centralization among institutions and low connection. The third category is the multivariate compact type, which is characterized by low network centrality and high network density, that is, low centralization and high connection between institutions. The fourth category is centralized and compact. Its performance is characterized by high network centrality and density, that is, high centralization and connection between institutions.

It can be seen from Table 3 that during SARS, the inter-agency network relationship of the Chinese public crisis governance system belongs to the second category, which is centralized and decentralized, with a network density of 0.063, a network center potential of 0.424, and an average distance of 1.930, representing the shortest average path for cooperation among various institutions. These data, together with network centrality, reflect the centralization degree of inter-agency network relations. During the period of COVID-19, the inter-agency network relationship of the Chinese public crisis governance system belonged to the third category, which is the multivariate compact type, and its network density is 0.098, which is higher than that during SARS, indicating that the inter-agency relationship of the Chinese public crisis governance system is increasing. During this period, the inter-agency network’s center potential was 0.392, which was a little lower than that during SARS, which indicated that the centralization degree between institutions in the Chinese public crisis governance system declined, and the whole network system changed to be multivariate in nature. Similarly, the average distance between network nodes during this period was 2.090, which was higher compared with that during SARS, also proving this change.

### 5.2. The Inter-Agency Network Relationship of the Chinese Public Crisis Governance System Is Expanding

As can be seen from Table 3, the number of policies during the period of COVID-19 was 455, which was a significant increase compared to 182 during the SARS period. Similarly, without considering institutional reform, the number of institutions participating in the Chinese public crisis governance system has also increased significantly: from 46 governance institutions during SARS to as many as 80 governance institutions during COVID-19. These data indicate that the inter-agency network relationship of the Chinese public crisis governance system is showing an expanding trend.

Comparing these two major epidemics, the increased institutions are mostly group organizations and industry associations, such as the China Disabled Persons’ Federation, the National Working Committee on Aging, the National Women’s Federation, and the National Patriotic Health Campaign Committee. The main reason for this is that the Chinese government pays more attention to the roles played by social organizations in the face of major epidemics. At the same time, banks and security institutions have also increased, such as the Export-Import Bank of China, the Shanghai Stock Exchange, Shenzhen Stock Exchange, etc. The increase in these institutions is mainly due to the long-lasting nature of the COVID-19 epidemic and its deep impact on the economy. There are also some external institutions showing an increasing trend, such as the Ministry of Foreign Affairs, the State Administration of Foreign Exchange, the State Administration of Immigration, and so on, which is mainly caused by the globalization of the COVID-19 epidemic.

On the whole, after the SARS and COVID-19 epidemics, Chinese institutional advantages are gradually changing to advantages of a governance system, concentrating more available forces and participating in crisis governance.

### 5.3. The Legal Governance of the Chinese Public Crisis Governance System Has Been Strengthened, and the Core of the Institutional Network Has Changed to the Leadership of the Party Committee

From Figure 2, it can be seen that the General Office of the CPC Central Committee and the General Office of the State Council jointly issued policies and did not participate in the overall network structure. However, this does not mean that these two publishing agencies are not important. On the contrary, as overall governance institutions, they represent global guidance for the entire governance system. In the same way, in Figure 3, the Central Committee of the Communist Party of China, the Central Military Commission, and the State Council also jointly issued a document independently of the overall network structure to make overall plans for the entire governance system and institutional network. In contrast, from the General Office of the CPC Central Committee to the Central Committee of the CPC, we find that the core of the institutional network of the Chinese public crisis governance system has changed. From indirect leadership by the Party Committee and the unified command by the government to direct leadership by the Party Committee and joint prevention and control by departments [26], the leadership of the Party has been continuously strengthened.

A more in-depth observation of the level and number of posts issued by agencies reveals that the number of posts issued by institutions with higher administrative levels such as the National Health Commission, the Ministry of Finance, the Ministry of Transport, and the National Development and Reform Commission increased during the COVID-19 period. Moreover, the Supreme People’s Procuratorate and the Supreme People’s Court also changed from a separate joint publication during SARS to participating in the overall network structure, and the relevant legal documents have also increased accordingly, which is also a necessity for China to adhere to the rule of law.

## 6. Conclusions

In these two 17-year-old epidemic crisis governance processes, each independent governance institution is in charge of its own corresponding governance scope, and these governance institutions play a vital role in their own fields. However, due to the publicity of public crises, these institutions cannot exist independently in the governance network. Therefore, they are connected with each other according to the relevance of crisis governance, forming a complex cooperative relationship. From the internal characteristics of public crisis, governance institutions with different functions encounter comprehensive events such as public crisis, especially when these events involve problems in various fields. It is an inevitable trend to unite and cooperate with each other. For example, the contact between the Ministry of Communications and the National Health Commission will be very close, and the two agencies will cooperate through the control of personnel flow and transmission to effectively control the epidemic situation. Despite the rapid development of the Chinese public crisis governance system, a new crisis situation may come at any time. Most of these crisis situations have no lessons learned from the past. In the face of a sudden crisis, it is obviously impossible to copy previous management methods. Due to the uncertainty and urgency of crisis events, a network governance structure with mutual cooperation between different functions and different jurisdictional areas has become the inevitable choice to solve crises.

### 6.1. The Network Relationship among Institutions in the Chinese Public Crisis Governance System Is in a Reasonable Evolution

From the perspective of the overall network structure, the institutions that occupy a more important role are mainly the departments of the State Council or agencies directly under the State Council and other ministries and commissions of similar levels. Moreover, compared with the two representative major epidemics, with the passage of time, due to the different impact scope and duration of crises as well as the institutional reform of the Chinese government, etc., the inter-agency network structure of the Chinese public crisis governance system will gradually evolve to rationalization. From the changes in low density and high network center potential during the period of SARS to the high density and low central potential during the COVID-19 outbreak, we can see that the network relationship among the institutions of the Chinese public crisis governance system has developed from a high degree of centralization and a low-degree network to a state of appropriate centralization and a high degree of network. Such an evolutionary process shows that the inter-agency relationship of the Chinese public crisis governance system is gradually developing to a reasonable network under the test of repeated unknown crises, preventing simple management of the original single government.

### 6.2. The Chinese Public Crisis Governance System Has Not Yet Formed a Cooperative Governance Structure of Coordination of Rights and Responsibilities, Optimization and Coordination, or One Core and Multiple Elements

After the two major epidemic crises of SARS and COVID-19, many changes have been observed in the Chinese public crisis governance system. Driven by the COVID-19 epidemic in particular, the Chinese public crisis governance system has become more perfect. However, in the public crisis governance system, cooperation is not only between the government and the government, but also between the government and social organizations. It also includes collaboration between the government and citizens. With the development of society, the uncertainty of public crises becomes more and more intense, and the complexity of crisis events also increases. In the face of these unknown difficult events, it is far from enough to rely on the network cooperation of government agencies. What we need to establish is a network structure system composed of three levels of multi-governance institutions: central–regional–local. Moreover, in each level of the crisis governance network, it is necessary to set up the core institutions for decision making at this level and to set up uncontroversial and clear institutional forms and functions. In this way, the cooperative actions among the member organizations in the hierarchical governance network are integrated [27]. In the process of policy formulation, the governance network needs to ensure a top-down policy formulation process to improve the response speed to public crisis. At the same time, due to the complexity of public crisis governance, the policy formation should strengthen discussion and coordination with experts from different domains including the colleagues responsible for on-site task execution. In this network system of governance institutions, while performing their own functions, institutions at all levels need to make all-round and three-dimensional contact with the vast number of social organizations and citizens to form a relaxed and well-organized network that covers all of society and that is ready to deal with unknown public crises at any time.

### 6.3. The Group Organization and Industry Association Have Not Fully Played Their Due Roles

During the COVID-19 outbreak, citizens have shown great differences in their emotional attitudes towards charitable organizations set up by individuals in Han Hong and the Wuhan Red Cross Society, which is mainly due to the accumulation of people’s distrust of government-run charitable organizations for many years [28]. This case shows that group organizations and industry associations still lack good communication with crisis network institutions and citizens. In the current public crisis governance system, compared with the SARS period, the role played by group organizations and industry associations has become more prominent, and the number of posts has increased, but the proportion in the overall network structure is still very small. The extensiveness of institutions involved in public crises and cooperative governance structures with one core or with multiple components is inevitable for future development. All organizations play an indispensable role. Therefore, in the future construction of a public crisis governance system in China, we must attach importance to the role of group organizations and industry associations.

To sum up, this paper analyzes the policies issued by institutions at the central level and statistically measures the density and central potential among institutions to further explore the interaction between institutions in the Chinese public crisis governance system, providing a new research method for the future research and improvement of the Chinese public crisis governance system.

However, this paper cannot completely and clearly explain the inter-agency relations within the public crisis governance system in China but hopes to provide some suitable research ideas for the future research on the related aspects of the public crisis governance system in China.

## Figures and Tables

**Figure 1 ijerph-19-15100-f001:**
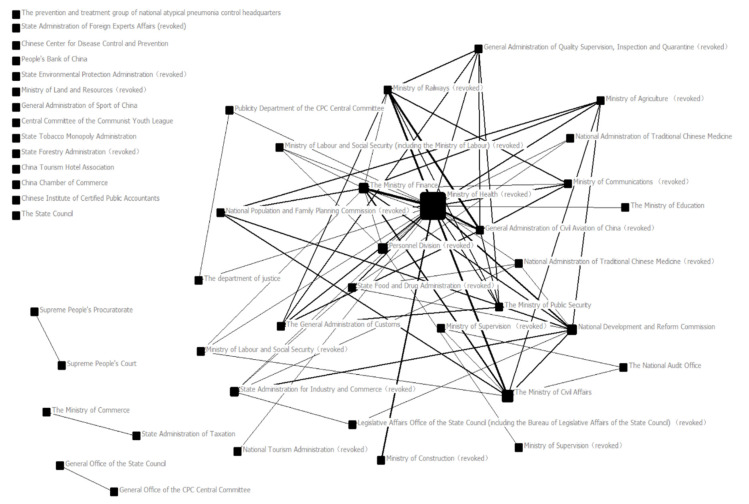
Inter-agency cooperation network for public policy making during SARS.

**Figure 2 ijerph-19-15100-f002:**
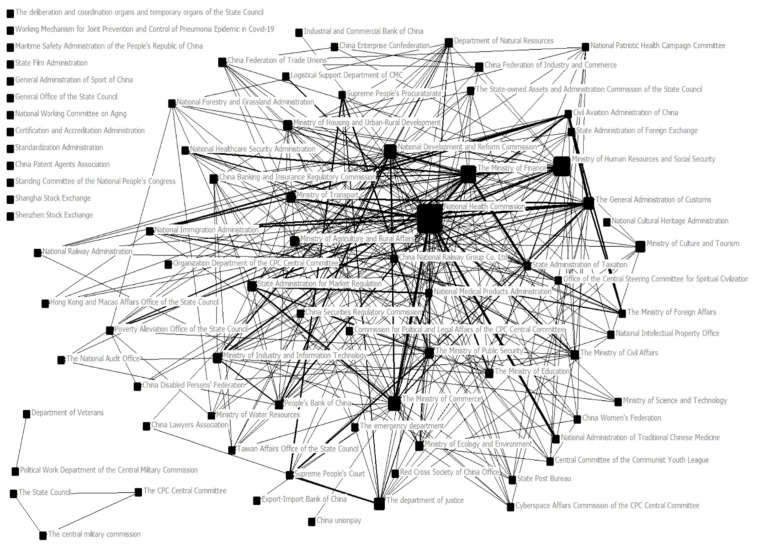
Inter-agency cooperation network of public policy making during COVID-19.

**Figure 3 ijerph-19-15100-f003:**
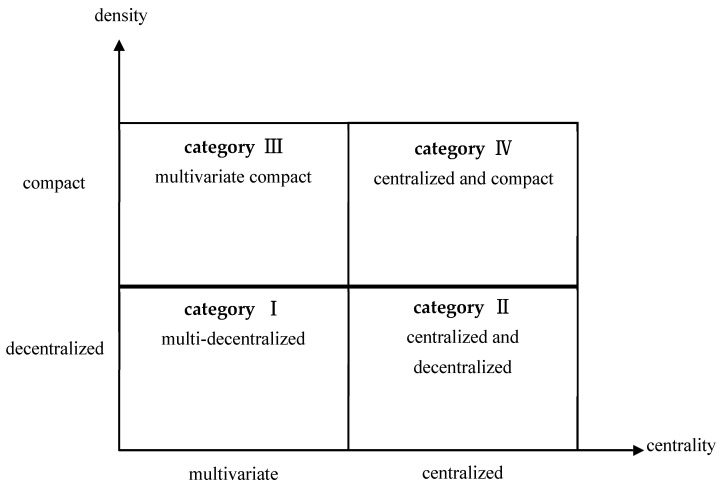
Types of institutional network relations in the public crisis governance system.

**Table 1 ijerph-19-15100-t001:** Policy distribution during SARS epidemic.

Issuing Agency	Issuing Level	Total Posts	Number of Issued Separately	Number of Issued Jointly
The State Council	Administrative laws and regulations	1	1	0
	Normative documents of the State Council	4	4	0
Party Central Agency, ministries and commissions of the State Council, directly affiliated agencies, and discussion and coordination agencies	Departmental regulations	2	2	0
	Departmental normative documents	98	90	8
	Departmental Working Paper	69	60	9
	Party regulations	2	0	2
Group organization	Group regulations	2	2	0
Industry association	Industry regulations	3	0	3
Supreme law and supreme inspection	Judicial interpretation	1	0	1

**Table 2 ijerph-19-15100-t002:** Policy distribution during the outbreak of COVID-19.

Issuing Agency	Issuing Level	Total Posts	Number of Issued Separately	Number of Issued Jointly
NPC Standing Committee	Law	1	1	0
The State Council, joint prevention and control mechanism	Administrative laws and regulations	1	0	1
	Normative documents of the State Council	19	19	0
Party Central Agency, ministries and commissions of the State Council, directly affiliated agencies, discussion and coordination agencies, and joint prevention and control mechanism	Departmental regulations	0	0	0
	Departmental normative documents	380	291	89
	Departmental Working Paper	33	20	13
	Party regulations	6	3	3
Group organization	Group regulations	5	3	2
Industry association	Industry regulations	2	2	0
Supreme law and supreme inspection	Work documents	1	0	1
	Judicial interpretation	7	4	3

**Table 3 ijerph-19-15100-t003:** Institutional characteristics of the inter-agency network of the Chinese public crisis governance system.

	During SARS (2 April 2003 to 30 December 2003)	During COVID-19 (23 January 2020 to 10 November 2020)
Scale (number of posts)	182	455
Number of nodes (issuing agency)	46	80
Network connection frequency	136	622
Central potential	0.424	0.392
Density	0.063	0.098
Average distance between nodes	1.930	2.090
Type of network structure	centralized and decentralized	multivariate compact

## Data Availability

Data are openly available in a public repository.

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
