# Peer review of "Institutional Network Relationship of Chinese Public Crisis Governance System—Based on the Quantitative Comparative Analysis of Policies during SARS and COVID-19"

_ijerph, 2022, doi:10.3390/ijerph192215100_

Round 1

Reviewer 1 Report

The Covid-19 pandemic has put the effectiveness of governments and the various institutions of every country in the world to the test. One of the important elements of crisis management is the institutional network relationships discussed in this article. Therefore, an interesting and topical issue has been taken up here.

In general, I believe that the article is prepared correctly, but there are a few details that should be improved.

I propose to precisely and comprehensively discuss the main goal and specific goals in one place (for example, at the end of section 3). In the present form, the partial objectives of the study are indicated in several places.

The drawings are illegible (1,2 !!!!) or show incomprehensible issues (3).

In conclusion, it is often said that public crisis governance system has become more perfect. On the basis of the conducted research, it cannot be assessed, because they do not concern the efficiency of the system, but only its structure.

These and other comments are discussed in detail in the work (in the comments).

Author Response

Point 1: I propose to precisely and comprehensively discuss the main goal and specific goals in one place (for example, at the end of section 3). In the present form, the partial objectives of the study are indicated in several places.

Response 1: We apologize for the unclear explanation of the goals. Accurately grasp the distribution structure of public crisis governance institutions and clarify the cooperation network structure in different periods have academic value and practical significance for the modernization of Chinese public crisis governance. This paper chooses the SARS and COVID-19 epidemics as the research focus to compare and analyze the internal changes in the mechanism of the inter-agency network relationship in the Chinese public crisis management system. This paper aims to enrich the knowledge of the relationship between Chinese governance institutions in the governance of major public health events. It also hopes to provide reference for other crisis governance in China and public health governance in other countries.

We merged the description of research objectives at the end of the Introduce part in line 79-97.

Point 2:The drawings are illegible (1,2 !!!!) or show incomprehensible issues (3).

Response 2:We apologize for the illegible drawings. The text data are processed by Ucinet6, and the central potential and density of cooperation networks of different institutions in different time periods are calculated. Netdraw software is used to deal with the matrix relationship between different publishing organizations, and the network maps between different governance organizations at the central level during SARS and COVID-19 are drawn in fig.1- fig.2. Each individual box in two network maps represents an independent governance institution, and the connection between two governance institutions means that there is joint publishing behavior between the two institutions. A thicker connection means that the two institutions are more closely connected; that is, in cases where there is more joint publishing, the size of each box is determined by intermediary centrality. This means that the number of shortest paths passing through a node, that is, the larger the box and the more frequent the connection between this institution and other institutions is, and this plays an intermediary role in the whole network structure. In this paper, by drawing and calculating the network map and network central potential during the two epidemics as well as by, combining the reform of government institutions in 2003 and 2018, we can conclude the characteristics and changes in network relations among institutions in the Chinese public crisis governance system.

Point 3:In conclusion, it is often said that public crisis governance system has become more perfect. On the basis of the conducted research, it cannot be assessed, because they do not concern the efficiency of the system, but only its structure.

Response 3:Thanks for pointing out this issue. This paper chooses the SARS and COVID-19 epidemics as the research focus to compare and analyze the internal changes in the mechanism of the inter-agency network relationship in the Chinese public crisis management system. This paper aims to enrich the knowledge of the relationship between Chinese governance institutions in the governance of major public health events. The efficiency of cooperation among the public health organization is not the focus of our research.

From the changes of low density and high network center potential during the period of SARS to the high density and low central potential during the COVID-19 outbreak, we can see that the network relationship among the institutions of the Chinese public crisis governance system has developed from a high degree of centralization and a low-degree network to a state of appropriate centralization and a high degree of network. This mainly shows that the links between departments are becoming closer, and the centralized nature is relatively weakened. From the perspective of policy coverage, the linkage and cooperation between various departments have been greatly improved. Therefore, it can be seen that the inter-agency network relationship of the Chinese public crisis governance system is improving and becoming reasonable, preventing simple management of the original single government.

The response to the comments are discussed in detail in the Pdf attachment.

Reviewer 2 Report

The area of the study requires attention from scholars and it would be good to sum up the literature in selected to the study area. The paper, so far, requires significant adjustments.

My detailed comments are as follows:

- the language must be significantly improved

- key words, please add "crisis management"

- the literature is to weak, please add considerably more positions

- there are almost no references at introduction section, whereas plenty of text should be supported with proper references. 

- please merge sections 2 and 3, consider shortening them and support with more references.

- please provide clearly the objective of the study at introduction section

- please separate discussion section from conclusions

- please think what are the most important conclusions from this study and provide them shortly in a separate section: conclusions

- please fill in lines 521-528

Author Response

Dear reviewer,

Thank you for your opinions, these comments are very helpful to improve the quality of the manuscript. We response the comments with a point by point and highlight the changes in revised manuscript. 

Reviewer 3 Report

In this paper, the authors examined the network relationship between institutions in the Chinese public crisis governance system from the two dimensions of network centrality and density. 

The goal of the research study shows the inter-activity of all organizations from viewpoints from the comparison of two public crisis cases. But, it appears lacks to discuss how to increase the efficiency of cooperation among the public health organization.

There are a number of details of the model that need more /explanation. First, it appears that a complete system architecture  of public health system is expected  and a detailed process of empirical experiment that describe how data were  analyzed. for example, the evaluation of network activities must describe including process of policy establishment, task implementation, task coordination and evaluation aspects. There is no explanation in the text.

I agreed that discussion 6.2.1. Chinese public crisis governance system has not yet formed a cooperative governance structure of coordination of rights and responsibilities, optimization and coordination.

Top-down policy formation may be needed to discuss and coordinate with experts from different domains including the colleagues who are responsible for the execution of tasks on site.

Author Response

(The authors gave the same response as above.)

Reviewer 4 Report

This study examines the network relationship between institutions in the Chinese public crisis governance system from the two dimensions of network centrality and network density during SARS and COVID-19. The work is in the scope of the journal, however, redaction and structure should be improved as indicated below, especially the methods should be clearer; the author is recommended to identify and practice sophisticated objectives for a journal publication. The author must justify the following points:

Comment 1: What is the scientific contribution of the presented work?

Comment 2: In the Introduction section, there is a lack of references for the several pieces of information that the author presented. Besides, a deep analysis of recent scientific papers covering only the topic and leading to the submission hypothesis based on the gap analysis of the previously published research is required.  

Comment 3: In line 55, what does “CNKI” stands for? Please check this issue for all other abbreviations in the whole manuscript.

Comment 4: The proposed approach of research methods is not outlined with the necessary vigor. The author needs to include sufficient methodological details in the paper and elaborate on the produced results from the proposed methods to justify the Inter-agency relations, Analytical framework, and Data sources. Some sections must be added and others need to be relocated and rewritten to make it clearer for the readers. This is an important issue to be justified and organized between sections 2, 3, and 4.  

Comment 5: Which program/software and database did the author use to simulate an inter-agency cooperation network for public policy making during SARS and COVID-19, as presented in Figures 1 and 2?

Comment 6: The Discussion Section should be presented right after the results, in an independent section, or integrated with the results section, before the conclusion section. The author needs to understand that this research is based on scientific questions. This section should be improved by including a clear and concise analysis of all results presented. You should analyze and comment on them more to indicate how you could achieve the aims of the study. It might be helpful to use more figures to present and discuss the results.

Comment 7: The Conclusion section is presented independently. This part of the study is missing some necessary details. For example, the author needs to highlight the novelty, aims, materials, and methods used in this work. Then the author should present the results of this work. Eventually, a summary of the limitations of this research as well as a recommendation for future works should be indicated.

Comment 8: Proofreading by a native English speaker should be conducted to improve clarity and organization quality. Besides, do not start with the title and subtitle without a text in between.

Author Response

(The authors gave the same response as above.)

Round 2

Reviewer 2 Report

The authors improved the paper.

Author Response

Dear reviewer,

It's a great honor to be recognized by you for our revision of this paper. Thanks again for your comments which are very helpful to improve the quality of our manuscript. 

Reviewer 4 Report

The work  has developed and the author answered my comments. One more thing that I would ask that do not start with the title and subtitle without a text in between. For example, between sections (2) and (2.1.), the author needs to add a text that explain what the reader will find in this section and how it has been divided. Please check this issue for the whole manuscript. 

Author Response

Dear reviewer,

Thanks again for your comments which are very helpful to improve the quality of our manuscript. We provide a point-by-point response (please see below) and highlight the changes in revised manuscript. 

Point:The work has developed and the author answered my comments. One more thing that I would ask that do not start with the title and subtitle without a text in between. For example, between sections (2) and (2.1.), the author needs to add a text that explain what the reader will find in this section and how it has been divided. Please check this issue for the whole manuscript.

Response:We are very sorry for our negligence. As you suggested, we have added texts between the title and subtitle in the whole manuscript, including between section2 and 2.1, section3 and 3.1, section4 and 4.1, section5 and 5.1, section6 and 6.1.